# Counteracting a Saturation Attack in Continuous-Variable Quantum Key Distribution Using an Adjustable Optical Filter Embedded in Homodyne Detector

**DOI:** 10.3390/e24030383

**Published:** 2022-03-09

**Authors:** Shengjie Xu, Yin Li, Yun Mao, Ying Guo

**Affiliations:** 1School of Automation, Central South University, Changsha 410083, China; 206214@csu.edu.cn (S.X.); liyin@csu.edu.cn (Y.L.); 2School of Computer, Beijing University of Posts and Telecommunications, Beijing 100876, China

**Keywords:** continuous-variable, quantum key distribution, saturation attack, adjustable optical filter

## Abstract

A saturation attack can be employed for compromising the practical security of continuous-variable quantum key distribution (CVQKD). In this paper, we suggest a countermeasure approach to resisting this attack by embedding an adjustable optical filter (AOF) in the CVQKD system. Numerical simulations illustrate the effects of the AOF-enabled countermeasure on the performance in terms of the secret key rate and transmission distance. The legal participants can trace back the information that has been eavesdropped by an attacker from the imperfect receiver, which indicates that this approach can be used for defeating a saturation attack in practical quantum communications.

## 1. Introduction

Quantum key distribution (QKD), which allows two legal parties, Alice and Bob, to share a set of secret key, may be manipulated by an eavesdropper, called Eve [1,2,3,4,5]. Currently, discrete-variable (DV) QKD has been developed, but it still faces challenges regarding the source preparation, the detection cost, and the secret key rate [6,7]. Continuous-variable (CV) QKD is another approach to actualizing QKD [8,9,10,11,12,13]. It has the advantage of convenient implementations, as it can be performed with diversification of sources such as coherent state [14] and squeezed state [15]. Nonetheless, CVQKD also faces threats of the practical security [16,17,18], resulting from device imperfections, technical deficiencies, and operational imperfections [10,19,20]. For example, Eve can perform a wavelength attack by controlling the transmittance of the wavelength-dependent beam splitter (BS) [21,22,23]. The calibration attack may be implemented by modifying the shape of the local oscillator (LO) pulse [24]. Consequently, several countermeasures have been proposed to counteract the effects of the LO calibration attack and the wavelength attacks [25,26,27].

In practical implementations of CVQKD, the coherent detector becomes vulnerable. Currently, the saturation attack has been performed while eavesdropping imperfect electronics in a homodyne detector [2,28]. It can be used for attacking the actual devices of the system, and thus it wakens the practical security because the coherent detector has a finite linearity domain that could be driven (if not being monitored) outside by displacing the mean value of the received quadratures. In addition, Eve may perform heterodyne detection to measure both quadratures *X* and *P* intercepted, and hence prepare for a faked coherent state [28,29]. In order to counteract such an attack, we may employ an embedded adjustable optical filter (AOF) in homodyne detectors that can be used to compensate for the potential saturation led by the strong received optical power in real time. The AOF-enabled detection, which is an actual gain adjustment of the avalanche photo-diode (APD), can be used for counteracting this saturation attack, based on the feedback of the response of detection.

This paper is organized as follows. In Section 2, we suggest an AOF-embedded CVQKD system to counteract the saturation attack. In Section 3, we perform numerical simulations to show effects of the AOF-enabled detection on the practical security of the CVQKD system. Finally, we conclude in Section 4.

## 2. The AOF-Embedded CVQKD

An eavesdropper can bias the excess noise estimation beyond the null key threshold by using the saturation attack, thus leading to a potential security loophole. In order to counteract this attack, an off-the-shelf detector has been employed at the receivers while performing data post-processing [28]. In this section, we consider an AOF-embedded CVQKD system that counteracts the saturation attack on-line, as shown in Figure 1a. The structure of the AOF-embedded CVQKD system is described in Appendix A. In addition, the AOF-enabled scheme is designed in Appendix B and the parameter estimation is derived in Appendix C, respectively.

The tunable AOF is employed for counteracting the saturation attack in CVQKD, where the data post-processing involves evaluation of attenuation ($\alpha_{\mathrm{tt}}$), which can be used for saturation compensation [28]. In Figure 1c, we illustrate the results of the saturation-involved attenuation evaluation, where abscissa $X_{A}$ is prepared for Alice, the ordinate $X_{Bsat}$ is Bob’s measurement results, and the red pots represent Eve’s measurement results. There are values of the saturation data $X_{\mathrm{sat}}^{\mathrm{Bsat}}$, the maximum data $X_{\max}^{A}$, and the saturation point $\left(X_{\mathrm{smin}}^{A},X_{\mathrm{sat}}^{\mathrm{Bsat}}\right)$, where $X_{\mathrm{smin}}^{A}$ is the minimum value sent by Alice when the measurement results are saturated. It is noted that $X_{\max}^{\mathrm{Bsat}}$ is the value corresponding to $X_{\max}^{A}$ in the blue line, which is derived by connecting the saturation point with the zero point. While making the measurement results in a finite linearity domain, we regulate the initial line from the black line after attenuation. Subsequently, we obtain the relationship of the blue line and the black line given by
(1)$$X_{\max}^{\mathrm{Bsat}}=k_{1}X_{\max}^{A},\;X_{\mathrm{sat}}^{\mathrm{Bsat}}=k_{2}X_{\max}^{A},$$
with the constraint $k_{2}=\alpha_{\mathrm{tt}}k_{1}$, where $\alpha_{\mathrm{tt}}$ is the attenuation with $\alpha_{\mathrm{tt}}=X_{\mathrm{sat}}^{\mathrm{Bsat}}{\left(X_{\max}^{\mathrm{Bsat}}\right)}^{-1}$. We note that $\alpha_{\mathrm{tt}}$ is an operation that should be performed at the receiver for data-processing with measurement results.

In what follows, we perform the data-processing for the operation $\alpha_{\mathrm{tt}}$, which is an algorithm for measurement results in essence. The initial attenuation $\alpha_{\mathrm{tt}}$ is assumed to be one. When the first data block is performed, the resulting attenuation $\alpha_{\mathrm{tt}}$ is updated on the initial one. The AOF is then performed for attenuation on the second data block according to the feedback of the previous attenuation. After that, the second block needs to derive the attenuation value. When there is no attenuation evaluated, the data block can be used to estimate the excess noise. Otherwise, the attenuation evaluated by the second data block is updated to attenuate the following block, and it has to repeat the aforementioned procedures.

## 3. Security Analysis

To demonstrate the effects of the AOF-enabled counteraction approach on the performance of system, we perform the saturation attack in CVQKD, which is illustrated in Appendix B. This strategy can be implemented by regulating the displacement Δ and the gain *g*. The effects of a saturation attack on parameter estimation are shown in Appendix C. We take into account measurements of data block size *N*, which is the number of coherent states prepared by Alice. In Figure 2, we show the effects of block size *N* on estimated excess noise with $N\in \{10^{6},10^{7},10^{8}\}$. We find that the large block size *N* may result in small excess noise. Without loss of generality, we consider numerical simulations of the AOF-embedded CVQKD system for $N=10^{7}$. In this section, all of the excess noises in numerical simulations are described in terms of shot-noise units.

### 3.1. Effects on Excess Noise

In Figure 3, we show measurement results under the saturation attack, where red dots, blue dots, and light dots denote measurement results for the saturation attack, the infinite linearity domain, and the attenuation, respectively. Due to the saturation attack, Alice and Bob may achieve the counterfeited information. However, as eavesdropping may increase the excess noise that make the generation of a secret key forbidden, Alice and Bob can detect the saturation attack in the traditional system, where the secret key rate may be decreased. In order to illustrate the effect of the AOF-enabled counteraction on the excess noise, we consider effects of the parameter displacement Δ*x* on the attacked CVQKD system. As shown in Figure 4, after performing the AOF-enabled counteraction scheme, the estimated excess noises fall in the finite linearity domain, which can lead to the performance improvement in terms of secret key rate.

### 3.2. Effects on the Secret Key Rate

The secret key rate using reverse reconciliation for the AOF-embedded CVQKD can be expressed as [28,29]
(2)$$K=\beta I_{\mathrm{AB}}-\chi_{\mathrm{BE}},$$
where β denotes the reconciliation efficiency, $I_{\mathrm{AB}}$ is the mutual information between Alice and Bob, and $\chi_{\mathrm{BE}}$ is the Holevo bound of Eve’s knowledge.

In Appendix C, we demonstrate the effects of parameters gain *g* and displacement Δx on the performance of the CVQKD system. Without loss of generality, we consider displacement Δx in numerical simulations. As shown in Figure 5a, Alice and Bob can extract the positive secret key rate when the transmission distance is more than 45 km. The large displacement Δx usually results in the long transmission distance. As Alice and Bob can achieve the positive secret key rate, Eve may succeed in stealing information without being discovered, leading to a security loophole. The reason is that when Eve performs the saturation attack, the secret key rate is positive, whereas the estimated excess noise is negative. However, after performing the AOF-enabled counteraction compensation, the secret key rate becomes negative, as shown in Figure 5b. As a consequence, Alice and Bob are able to detect the potential eavesdropper since there is no secret key generated from the resulting system.

## 4. Conclusions

We have proposed an AOF-embedded CVQKD to resist the saturation attack for performance improvement of the practical security. The numerical simulations show that after performing the AOF in the linear domain, the estimated excess noise is made more than zero, and the secret key rate is less than zero. The legal participants can detect Eve, who performs the saturation attack. Based on the AOF-enabled countermeasure compensation, the saturation attack can be broken to enhance the practical security, which provides a useful approach to increasing the practical security of the CVQKD system. In addition, there might exist other approaches for counteracting a saturation attack, such as the self-adapting detection of the eavesdroppers with machine learning or deep learning, which will be investigated in our future work.

## Figures and Tables

**Figure 1 entropy-24-00383-f001:**
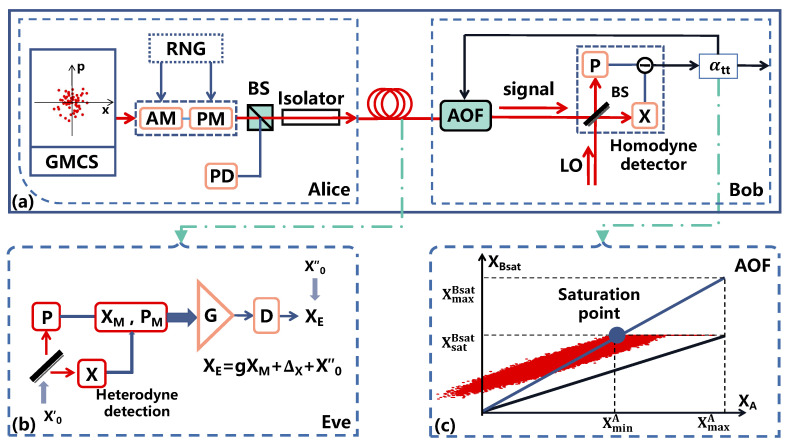
The AOF-embedded CVQKD. (**a**) System diagram: Alice prepares for coherent states with quadratures *X* and *P*; Bob performs homodyne detection; AM, amplitude modulator; PM, phase modulator; BS, beam splitter; PD, photodetector. (**b**) Strategy of saturation attack on CVQKD. G, gain *g*; D, displacement Δ. (**c**) Demonstration of operation $\alpha_{\mathrm{tt}}$ for AOF.

**Figure 2 entropy-24-00383-f002:**
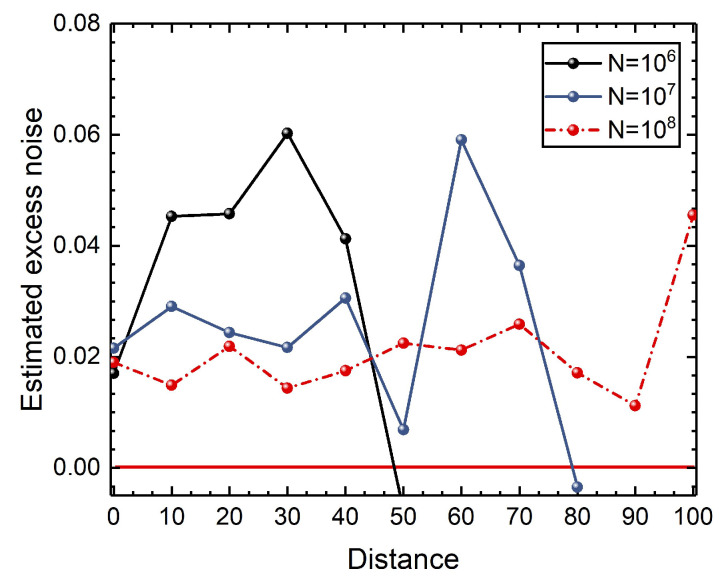
Effects of block size *N* on the estimated excess noise. The excess noise in numerical simulations are described in terms of shot-noise units.

**Figure 3 entropy-24-00383-f003:**
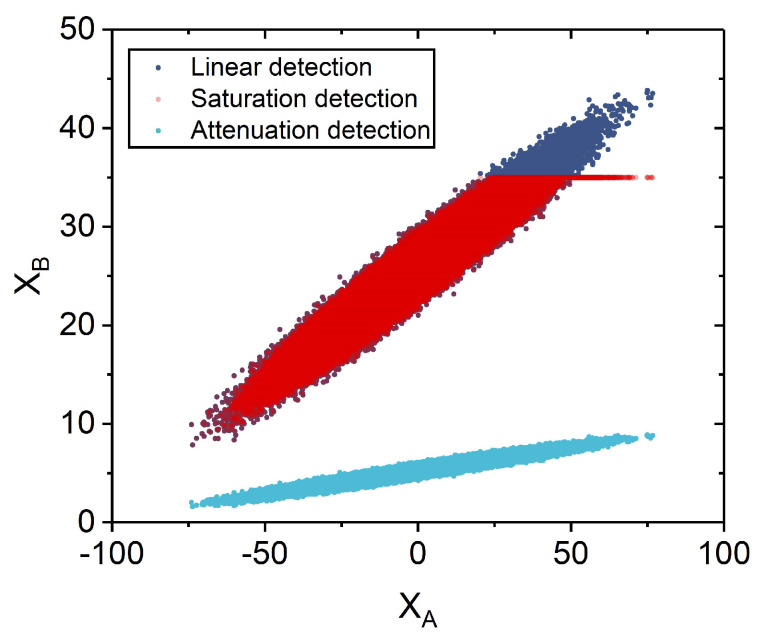
Measurement results. Red dots: results under the saturation attack; Blue dots: results under saturation attack; Light blue dots: results after being attenuated. Experimental parameters: $X_{\mathrm{sat}}^{\mathrm{Bsat}}=35\sqrt{N_{0}}$, Δx=110, and $N=10^{7}$.

**Figure 4 entropy-24-00383-f004:**
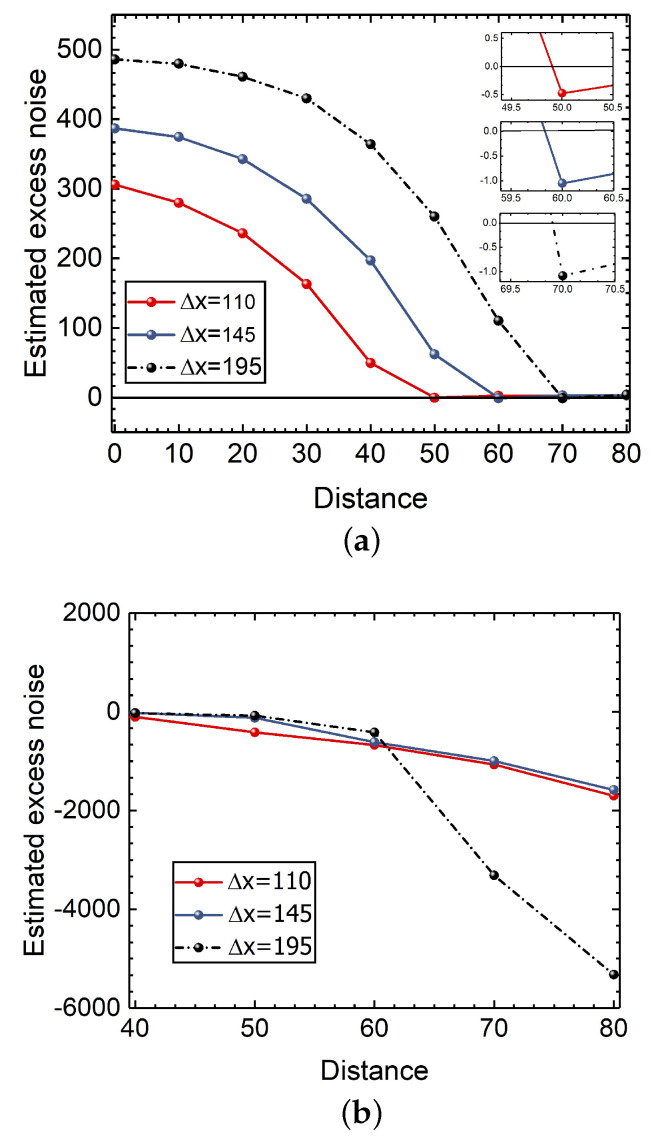
The estimated excess noise of the CVQKD system. (**a**) The traditional system under saturation attack. (**b**) The AOF-embedded system under saturation attack.

**Figure 5 entropy-24-00383-f005:**
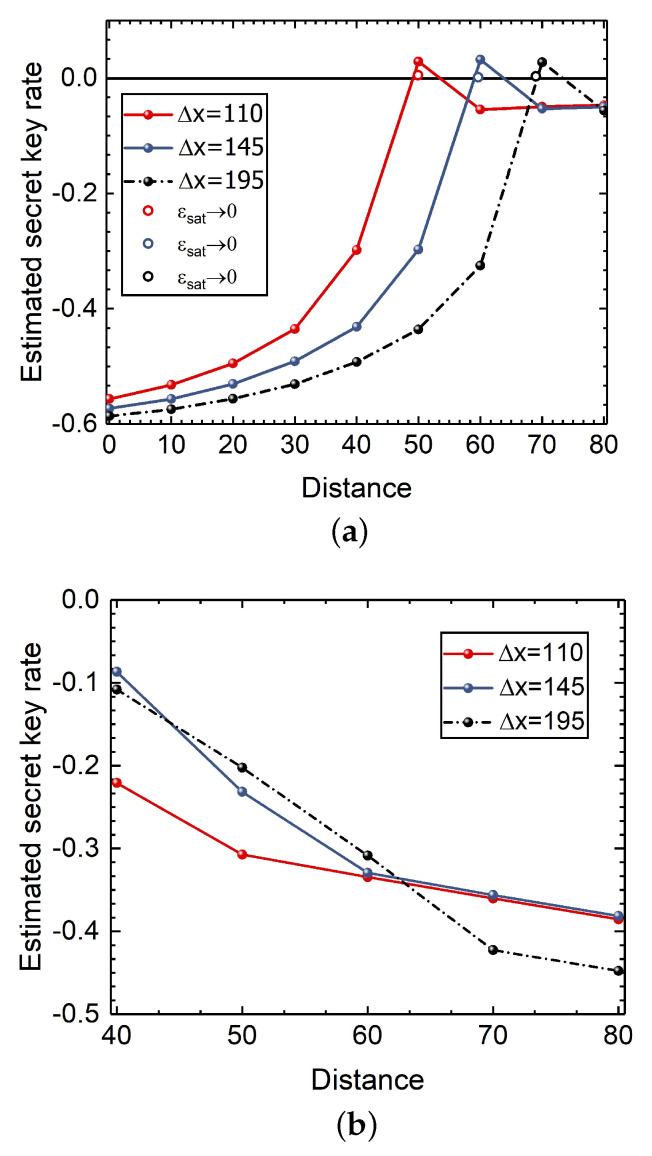
The secret key rate of the CVQKD system. (**a**) The secret key rate of the traditional system under saturation attack. The hollow dots represent the value evaluated by the excess noise approaching to zero. (**b**) The secret key rate of the AOF-embedded system.

## Data Availability

Not applicable.

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
