# Peer review of "Counteracting a Saturation Attack in Continuous-Variable Quantum Key Distribution Using an Adjustable Optical Filter Embedded in Homodyne Detector"

_entropy, 2022, doi:10.3390/e24030383_

Round 1

Reviewer 1 Report

In this work an AOF-embedded CVQKD to resist the saturation attack in terms of the practical security is proposed. The topic is important in the field of CVQKD and the paper’s proposal deserves dissemination. 

Nevertheless, I have some suggestions that may enhance the readability and widen the paper’s scope.

1.- It would be helpful for the unfamiliar reader a short explanation of the basics of CVQKD. For example, where is the message.

2.- I think that the saturation attack by Eve should be more clearly explained. In particular, the deviation of the linearity domain by Eve.

3.- Brief mention of other proposals of counteracting saturation attack on CVQKD, like that on ref. 9, or any other.

4.- The excess noise in the figures has no units.

5.- There are some typos.

If the authors accomplish these suggestions, I recommend acceptance.

Author Response

Thank you very much for your careful and constructive comments. Those comments are all valuable and very helpful for revising and improving our paper, and with the important guiding significance to us. We have tried our best to revise and improve the manuscript and made great changes in the manuscript. Please refer to the revised paper and following answers. 

Comment 0: In this work an AOF-embedded CVQKD to resist the saturation attack in terms of the practical security is proposed. The topic is important in the CVQKD and the paper’s proposal deserves dissemination. Nevertheless, I have some suggestions that may enhance the readability and widen the paper’s scope.

Comment 1: I think that the saturation attack by Eve should be more clearly explained. In particular, the deviation of the linearity domain by Eve.

Response: We appreciate reviewer's careful review. As for the the saturation attack by Eve at the receiver, which may bring out the practical security for the implementation of the CVQKD system, has been suggested several researchers. The attacking strategy has been described in Figure 1(b). In addition, in Ref [29], Liu W, et al demonstrated the feasibility of the saturation attack in CVQKD and suggested a compensation approach for visible light communication with off-the-shelf detectors[Optics Express 2021, 29, 9670-9684]. This reference has been cited in the manuscript. In the revised version, we have detailed the Eve’s attacking in Appendix B, as shown on Page 9.

Comment 2: Brief mention of other proposals of counteracting saturation attack on CVQKD, like that on ref. 9, or any other.

Response: There are several schemes that have been suggested for counteracting saturation attack on CVQKD. As proposals of counteracting saturation attack on CVQKD, we have suggested it in Appendix B and Appendix C on Page 9. We noted in the revised version,  that off-the-shelf detectors [29] can be used for counteracting this saturation attack. Please see the first paragraph in Section 2 on Page 2. In addition, the machine learning-based data-processing can be potentially employed for counteracting saturation attack, which will be considered in our future work. Please see the revised conclusion in Section 4 on Page 4.

Comment 3: The excess noise in the figures has no units.

Response: We are sorry that we have ignored the description of excess noise. We note that all of the excess noise in the related numerical simulations are described in terms of the shot-noise units. This assumption, which has been actually implemented for our previous work, has been used for all figures in this manuscript. Please see the notation in the first paragraph in Section 2 on Page 2. In addition, we note this description in Figure 2.

Comment 4: There are some typos.

Response: Thank you so much for your good comments and helpful suggestions, which can make our paper more comprehensive. Regarding the English language, we have checked and revised completely.

We have tried our best to revise and improve the manuscript and made great changes in the manuscript according to the reviewers’ constructive comments, and we hope that the corrections will meet with approval. Once again, thank you very much for thoughtful and constructive comments that greatly help to improve the results and presentation of this paper.

Reviewer 2 Report

In this paper “Counteracting saturation attack with adjustable optical filter embedded in continuous-variable quantum key distribution” authors propose a method to detect the presence of an eavesdropper in a saturation attack to a CV-QKD system.

Authors start by describing the method proposed and how it can incorporated in a CV-QKD system, then they analyze the impact of this solution to the security of the CV-QKD with particular focus to channel excess noise. At the end of the manuscript they present the conclusions of their work.

The paper is well written, and the state of the art is clearly presented. Besides that, the novelty is clearly presented.

Considering the state of the art, the study presented in this work could be interesting to the scientific community, in my opinion this paper should be improved before being accepted for publication.

Comments to authors:

  • The description of the figures on this manuscript are very poor. Authors should improve the description of their results.
  • Authors mention on the manuscript that they use measured results. My concern is the following: Authors implemented in the laboratory a CV-QKD system? That is unclear from a careful read of the manuscript. A detailed description of the setup of Fig. 1 must be done.
  • Besides that, why authors focus on homodyne detection? Nevertheless, most of the works related with CV-QKD systems are nowadays focused on heterodyne detection since it can measure simultaneously both quadratures. Why authors do not use heterodyne detection since it increases the secure key rate?

In my opinion this is an essential point to bring novelty to the manuscript considering the state or the art (Qin H, Kumar R, Allaume R. Quantum hacking: Saturation attack on practical continuous-variable quantum key distribution. Physical Review A, 2016, 94,012325)

  • When studying security, two parameters must be precisely estimated: the channel transmissivity; the channel excess noise. In this work authors focus only on channel excess noise. Why authors do not present results for the channels transmissivity and the impact of the AOF on its estimation?
  • Another concern is: to achieve security it is essential to precisely estimate the system shot-noise. Ideally, shot noise have infinity bandwidth (electric and/or optical). How the presence of an AOF can lead to an incorrect estimation of Bob shot noise and consequently to an over or under estimation of secret key rate?
  • Can authors clarify in the manuscript the error obtained through the estimation of excess noise?
  • Finally, and must important authors should clarify in the manuscript how the AOF can be implemented in practice and its specifications for being used on the CV-QKD system. Moreover the parameter \alpha_{tt} in fig. 1 must be also better described and how it arises from the measurement and how it is used on the AOF.

Author Response

Thank you very much for your careful and constructive comments. Those comments are all valuable and very helpful for revising and improving our paper, and with the important guiding significance to us. We have tried our best to revise and improve the manuscript and made great changes in the manuscript. Please refer to the revised paper and following answers.

Comment 1: The description of the figures on this manuscript are very poor. Authors should improve the description of their results.

Response: Thanks to your comprehensive consideration about presentation of our manuscript. We have revised carefully. In Figure 1 and Figure 2, we demonstrate the characteristics of the AOF-based CVQKD. The detailed description has been shown in Appendix A, please see them in the revised version. For the figures in numerical simulations, since we focused on scheme design for counteracting saturation attack on CVQKD, we only demonstrate the structure of the CVQKD system without considering the practical performance of the constructed system, and thus have revised them carefully where they are required.

Comment 2: Authors mention on the manuscript that they use measured results. My concern is the following: Authors implemented in the laboratory a CV-QKD system? That is unclear from a careful read of the manuscript. A detailed description of the setup of Fig. 1 must be done.

Response: We thank a lots for this comments. We are sorry that we focused on the scheme design for the performance improvement in terms of the practical security. We note that all of the performance analysis have been done in numerical simulations. As for its implementations, it will be done in our future work. In addition, we are sorry that we have ignored the detailed description of the Figure 1. Therefore, we have added the detailed CVQKD scheme and AOF-based CVQKD in Appendix A and Appendix B. The parameter estimation of the related system in shown in Appendix C.

Comment 3: In my opinion this is an essential point to bring novelty to the manuscript considering the state or the art (Qin H, Kumar R, Allaume R. Quantum hacking: Saturation attack on practical continuous-variable quantum key distribution. Physical Review A, 2016, 94,012325)

Response: We note that the contribution of our work is to suggest an approach to counteract saturation attack on CVQKD, which has bee suggested by Qin H et al [Physical Review A, 2016, 94,012325]. Though several approaches has been suggested [Optics Express 2021, 29, 9670-9684], we yet proposed another approach, which is an AOF-enabled counteracting scheme, to counteract saturation attack. This approach can be done on-line before performing data post-processing.

Comment 4: When studying security, two parameters must be precisely estimated: the channel transmissivity and the channel excess noise. In this work authors focus only on channel excess noise. Why authors do not present results for the channels transmissivity and the impact of the AOF on its estimation?

Response: As shown in Appendix C, When studying two parameters of the related CVQKD system, the channel transmissivity and the channel excess noise should be estimated. The two parameters have also been derived in Appendix C. Because the aim of this paper is to show the AOF-based counteracting scheme against the saturation attack, we only consider effect of AOF on the channel excess noise while ignoring effect of AOF on the channel transmissivity without loss of generality. In addition, in order to reply this commend, we demonstrate the effect of AOF on transmissivity in Appendix C. Actually, we can similarly consider the effect on the channel transmissivity since we have derive its description in formula. Pleas see Eq. (3) and Eq. (7) on Page 5.

Comment 5: Besides that, why authors focus on homodyne detection? Nevertheless, most of the works related with CV-QKD systems are nowadays focused on heterodyne detection since it can measure simultaneously both quadratures. Why authors do not use heterodyne detection since it increases the secure key rate?

Response: We are agree with this nice suggest. It is well known that while performing with heterodyne detection the performance of the CVQKD system in terms of secret key rate is higher than that of secret key rate with homodyne detection. However, when considering the maximal transmission distance, it demonstrates the bad performance contribution. Therefore, we can also make use of heterodyne detection to show the effect of AOF on the performance of the CVQKD system. Since the aim of this paper is to compare with the performances of the AOF-enabled counteracting scheme and the tradition CVQKD system, we can achieve the same results when using  heterodyne detection. Namely, even when using heterodyne detection, we can draw the same conclusion that the AOF-based CVQKD can demonstrate the good performance when counteracting the saturation attack in CVQKD. As for the difference of homodyne detection and heterodyne detection, please see it in Appendix A on Page 8.

Comment 6: Another concern is: to achieve security it is essential to precisely estimate the system shot-noise. Ideally, shot noise have infinity bandwidth (electric and/or optical). How the presence of an AOF can lead to an incorrect estimation of Bob shot noise and consequently to an over or under estimation of secret key rate? Can authors clarify in the manuscript the error obtained through the estimation of excess noise?

Response: As shown in numerical simulations while demonstrating effect of the AOF-enabled scheme on performance of the CVQKD system in terms of excess noise, it is described in the shot-noise unit. Although the AOF scheme can counteract the saturation attack in CVQKD, it may lead to an incorrect estimation of Bob shot noise and consequently to an under estimation of secret key rate in practice. This is an shortcoming of this work. Using the parameter estimations in Appendix C, we can derive the the error obtained through the estimation of excess noise. The error estimation is an important parameter that we should be mentioned, however it is out of this work and we will consider this parameter in the future. 

Comment 7: Finally, and must important authors should clarify in the manuscript how the AOF can be implemented in practice and its specifications for being used on the CV-QKD system. Moreover the parameter \alpha_{tt} in fig. 1 must be also better described and how it arises from the measurement and how it is used on the AOF.

Response: We are sorry that we have ignored the implementation of the AOF in the CV-QKD system. Consequently, we have added this implementation in Appendix. In detail, please see Appendix C. Moreover, as shown in Eq.(1), the parameter \alpha_{tt} can be derived by the relation of X_{\rm {max}}^{{\rm {Bsat}}} and X_{\rm {sat}}^{{\rm {Bsat}}}. Actually, the parameter \alpha_{tt} is an operation that can be used for resisting the saturation attack. This action can be performed with the algorithm at the receiver. The operation \alpha_{tt} is performed on the basis of the feedback of the response of  detection that results in saturation. The \alpha_{tt}-related data processing is described in the last paragraph in Section 2 on Page 2.   

We have tried our best to revise and improve the manuscript and made great changes in the manuscript according to the reviewers’ constructive comments, and we hope that the corrections will meet with approval. Once again, thank you very much for thoughtful and constructive comments that greatly help to improve the results and presentation of this paper.

Round 2

Reviewer 2 Report

No further comments regarding the response to authors respononse to my comments